# Is Memorization *Actually* Necessary for Generalization?

## Abstract

Memorization is the ability of deep models to associate training data with seemingly random labels. Even though memorization may not align with a model's ability to generalize, recent work by Feldman & Zhang (2020) has demonstrated that memorization is in fact *necessary* for generalization. However, upon closer inspection, we find that their methodology has three limitations. First, the definition of memorization is imprecise, leading to contradictory results. Second, their proposed algorithm used for *approximating* the leave-one-out test (the gold standard for calculating memorization scores) suffers from a high approximation error. Three, the authors induce a distribution shift when calculating marginal utility, leading to flawed results. Having accounted for these errors, we re-evaluate the role of memorization on generalization. To do so, we track how memorization changes at different levels of generalization (test accuracy). We control model generalization by training 19 different combinations of models, datasets, and training optimizations. We find that memorization and generalization are *strongly* negatively correlated (Pearson -0.997): As one decreases, the other increases. This shows that memorization is not necessary for generalization, as otherwise, the correlation would have been positive. In light of these findings, future researchers are encouraged to design techniques that can accurately approximate memorization scores.

## 1 Introduction

One of the most interesting properties of deep learning models is their ability to fit outliers (i.e., samples that are not part of the data distribution) (Zhang et al., 2017a; Arpit et al., 2017; Stephenson et al., 2021). Specifically, deep models can output arbitrary ground-truth labels to inputs in the data set. For example, if a picture of Gaussian noise is mislabeled as a cat, then the model will output this label, even though the label is incorrect (Zhang et al., 2017a). This is only possible due to the model's ability to *memorize* point-label pairs.

Intuitively, the ability to generalize (i.e., correctly label previously unseen points) should be at odds with memorization. This is because generalization requires identifying the underlying patterns and then subsequently applying them to unseen points. On the other hand, memorization simply retrieves the labels of the previously observed inputs and consequently, should not help in correctly classifying new unseen points. However, recent work from Feldman & Zhang (2020) has shown that this is not true for deep models. Their work demonstrated that "memorization is necessary for achieving close-to-optimal generalization error". They do so using a three-step process. **First**, they provide a definition that quantifies a given point's memorization score (i.e., the likelihood that the point is memorized). **Second**, they propose an approximation method to calculate memorization scores for each point in the data set. **Third**, they calculate the marginal utility of memorization (i.e., its impact on test set accuracy). They report an accuracy degradation of 2.54 ± 0.20%, thereby concluding that memorization is necessary for generalization. However, their work has three core issues:

1. **Imprecise Definition:** We find that the definition of memorization, which also forms the basis of their theoretical work, is imprecise. It does not provide any standard threshold to identify memorized points: any point above an arbitrary threshold is considered memorized. We find that different threshold values capture different sets of points, with high thresholds only capturing the singleton outliers, and lower thresholds capturing entire or

partial sub-populations. As a result, two different thresholds can lead to contradictory conclusions. In other words, experiments under this definition are not falsifiable, a basic tenant of scientific research. To mitigate this issue, we evaluate marginal utility over multiple thresholds and base our conclusions on consistent behavior over most threshold values.

2. **High Approximation Error:** We find that Feldman (2020)'s approximation algorithm suffers from a high error rate, significantly overestimating the memorization scores. When we compare its scores against the baseline leave-one-out experiment, we found that: 1) only a small fraction of the points (less than 6%) were within 5% error, 2) the remaining points have an exceptionally high error, as much as, 50%, 3) resulting in a high rate of false positives, with as many as 93% of the points being incorrectly marked as memorized. Upon closer inspection, we find that small sub-populations are most vulnerable to these errors. As a result, the approximation algorithm incorrectly groups them with the actual memorized points, even though the small sub-populations are not memorized. To ameliorate this limitation, we identify and remove the points with incorrect scores from the memorized set.

3. **Flawed Marginal Utility:** The authors removed the memorized points from the data to calculate their marginal utility. However, in doing so, the authors accidentally purge entire sub-populations, wholesale. This leads to a distribution shift (i.e., loss of sub-population), a common oversight in ML research. As a result, we find the drop in accuracy was not a product of the marginal utility of memorized points, but due to common ML oversight.

**Re-evaluating Memorization vs Generalization:** One simple way to fix these issues to take a different approach to study memorization and generalization. Instead of measuring the drop in accuracy after removing memorized points, we track how memorization changes at different levels of generalization (test accuracy). We control model generalization by training 19 different combinations of models, datasets, and training optimizations. Specifically, we vary test accuracy by: 1) changing model complexity, (number of trainable weights), 2) changing training optimizations (With and without weight decay and data augmentation), 3) across different models (VGG, ResNet, and ViT) and 4) across datasets (Cifar-10/100 and Tiny ImageNet).

We then plot the corresponding rates of memorization and generalization. We find that there is a *strong negative* correlation between memorization and generalization (Pearson -0.997). This means that as generalization improves, memorization decreases (and vice versa). From the results of our experiments, we disprove the conclusion of the original work and show that *memorization is not necessary for generalization*.

## 2 BACKGROUND

Before we discuss our findings in any detail, it is important that we first understand some of the important background concepts regarding memorization.

### 2.1 SUB-POPULATIONS

A data set can consist of one or more coarse class labels (e.g., cats and dogs). Within each of these coarse labels, there may exist a mixture of points that have finer labels, which correspond to distinct sub-populations (Zhu et al., 2014). For example, the cat data set will contain images with different cat features including color, background, species, pose, etc. Cats with the same facets will fall into the same sub-populations. For example, consider a hypothetical data set that contains 100 cat images, with 94 white cats, three black, and a single pink one. Even though they have the same label, each color of cats forms distinct sub-populations (with potentially even finer sub-populations within the white and black cats respectively (Malisiewicz et al., 2011; Felzenszwalb et al., 2009)). Figure 1 provides examples of sub-populations in CIFAR-100[1].

The size of the sub-population may impact model accuracy as well. Generally, the larger the sub-population, the higher the number of exemplar points, and the greater the model's ability to predict accurately for that sub-population at test time Jiang et al. (2020). This is because more points usually mean more representative examples for the model to learn from. Returning to our hypothetical cat data set (with 94 white cats, three black, and a single pink one), since there are more pictures of white

---

[1]In Section 4 we describe how we find the sub-populations within a dataset.

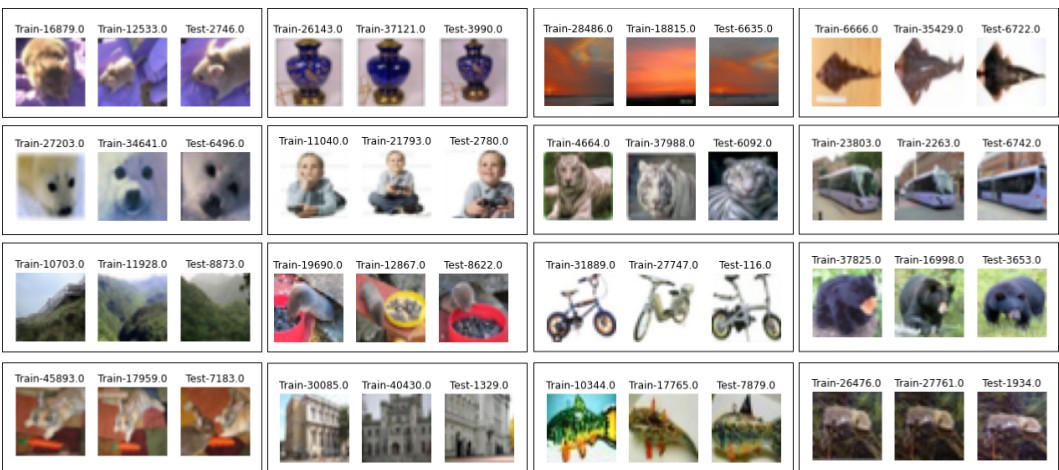

Figure 1: Examples of sub-populations that suffer from the highest approximation error.

cats in the data set, the model likely has better prediction accuracy on white cats than black ones at test time. This also means removing an entire sub-population will result in degrading the model's ability to correctly classify missing sub-populations at test time. If the model was never trained on black cats, it will likely misclassify them at a higher rate. This is because certain distinguishing characteristics that aid the model in correctly classifying the unseen sub-population will likely not be learned from remaining data, and will negatively impact model performance on unseen points.

## 2.2 INFLUENCE OF DATA SET POINTS

Influence is the ability of a training point $(x_i, y_i)$ to impact the prediction of some other point $(x_j, y_j)$ (Koh & Liang, 2017). Hereinafter, $x_i$ denotes the data point and $y_i$ denotes the label of $x_i$. As an illustrative example for influence, if including $x_i$ in the training set increases the probability of point $x_j$ being classified correctly by the resulting model, then $x_i$ is said to have a *positive* influence on $x_j$ (Feldman & Zhang, 2020). The higher the influence value, the stronger the impact.

*Self-influence* is a special case of influence. It is a measure of how well the model predicts the label for $x_i$ when the point itself is present in the training data set in comparison to when $x_i$ is absent. If a point has positive self-influence, it has a higher probability of being classified correctly when it is present in the training data set. Therefore, when the point is removed from the training data set, the likelihood of correct prediction goes down as well. Conversely, negative self-influence means a higher likelihood of being classified correctly only if it is not present in the training data set (e.g., mislabeled points).

According to Feldman & Zhang (2020), higher self-influence means a higher risk of memorization. The high self-influence points usually belong to the tail end of the data distribution. The tail usually consists of atypical points (e.g., outliers and mislabeled points) or small-sized sub-populations (e.g., five black cats in a data set of all white cats). Therefore, these points have the highest risk of memorization across the entire distribution.

Furthermore, if the point has high self-influence *and* has a duplicate in the test set, then removing this point from the training data will result in the wrong prediction on itself, but also its duplicate (or near duplicate) in the test set.

## 3 UNDERSTANDING FELDMAN & ZHANG (2020)

Having gone over how different factors influence memorization, we describe in detail the original work of Feldman & Zhang (2020). Our primary goal is to evaluate their methodology, recommend experimental fixes, and consequently, reassess their findings. To that end, we describe how they 1) define memorization, 2) approximate memorization scores, and 3) quantify marginal utility.

## 3.1 Defining Memorization

Feldman & Zhang (2020) define a memorized point as one having high self-influence (i.e., a point that is predicted correctly only when present in the training data).

Specifically, consider a training set $S = ((x_1, y_1)...(x_n, y_n))$ and a point $x_i$ in the training set $S$. The memorization score is the difference in prediction accuracy between when the point $x_i$ is present in the training data ($h \leftarrow A(S)$) and when $x_i$ is absent ($h \leftarrow A(S^{\setminus i})$). Here, ($h \leftarrow A(S)$) means that models $h$ were trained on dataset $S$ using algorithm $A$:

$$\mathbf{Pr}_{h \leftarrow A(S)}[h(x_i) = y_i] - \mathbf{Pr}_{h \leftarrow A(S^{\setminus i})}[h(x_i) = y_i] \tag{1}$$

The definition captures the intuition that a point $x_i$ has been memorized if its prediction changes significantly when it is removed from the dataset. We include Table 1 for reference on the symbols used throughout the paper.

For example, consider training 1000 instances each of the models $h \leftarrow A(S)$ and $h \leftarrow A(S^{\setminus i})$. If the correct classification rate for $x_i$ when it $h \leftarrow A(S)$ is around 90% (i.e., 900 out of the 1000 instances classified the point correctly). However, it falls significantly when $h \leftarrow A(S^{\setminus i})$ to 25% (i.e., 250 out of the 1000 instances classified the point correctly). Due to the significant drop in self accuracy, this point has a high self-influence, and therefore, a high memorization score, specifically of $90\% - 25\% = 65\%$. This means that $x_i$ is far more likely be classified correctly when it is present in the training data. In contrast, if there is no significant change in the classification rate, then it has a low memorization score. In this case, $x_i$ will likely be classified correctly, whether or not it is present in the train set. As a result, the memorization score of a given point will be inversely proportional to its sub-population size: the larger the sub-population the smaller the memorization score. In the case of our hypothetical cat dataset, the pink cat (singleton) will have the highest memorization score, followed by the black cats (small-subpopulation) and the white cats (large sub-population).

## 3.2 Calculating Memorization Scores

**Precise Algorithm:** Having defined memorization, the next step is to develop a methodology to identify memorized points from a dataset. A point is considered memorized based on its memorization score, calculated using Equation 1. The most *precise* way to compute this score is via the classic leave-one-out experiment. Here, we remove a single point from the training dataset, retrain the model on the remaining data, and test to see if the removed point is correctly classified. We have to run this experiment on all the points in the dataset to get the memorization score for each. Additionally, we have to repeat this model training process, for each point, multiple times to account for randomness introduced during training (e.g., the varying initialization, GPU randomness, etc.). Specifically, this would require training hundreds models for every point in the data. Considering data sets contain tens of thousands of points, this would require training millions of models. Therefore, running this experiment over a large dataset and model will require a large amount of resources and is therefore, computationally intractable.

**Approximation Algorithm:** To overcome this limitation, Feldman & Zhang (2020) propose a method to *approximate* the memorization scores. Instead of removing one point at a time, the authors randomly sample a fraction $r$ of the points from the training set (originally of size $n$) and leave the remaining points out of training. The number of points used in training is then $m = r \cdot n$, $0 \leq r \leq 1$. In Feldman & Zhang (2020) the authors use $r = 0.7$ for their experiments. The authors repeat this $k$ times. The exact value of $k$ depends on the dataset but is typically on the order of a few thousand models. As a result, a random point $x_i$ will be present in approximately $k \cdot r$ of the total trained models and will be absent from $k \cdot (1 - r)$ of them. By aggregating the results over both sets of models, the authors can approximate the memorization score for $x_i$. All the points that have a higher memorization score than some predetermined threshold are said to be memorized.

## 3.3 Calculating Marginal Utility

Having identified the memorized points, the authors now calculate their marginal utility (i.e., their impact on test accuracy). This is done using a two-step process:

**Step 1: Training Models without the memorized points** The authors train two sets of models: one on the full training data (that includes the memorized points), and another on the reduced dataset

(without the memorized points). They train both sets of models on identical parameters, repeating this procedure hundreds of times to account for sources of randomness. At this point, the authors have hundreds of models trained on the full and reduced datasets.

**Step 2: Measuring the Difference in Accuracy** Next, the authors measure the drop in accuracy caused by removing the memorized points and retraining the models. They simply take the mean test set accuracy of the models trained on the full dataset and the models trained on the reduced one respectively. They subtract the two accuracies to find the mean difference and the standard deviation. The authors reported a significant drop in accuracy of 2.54 ± 0.20% and therefore, the concluded that these memorized points need to be present in the training data for optimum accuracy. And as a result, memorization is necessary for generalization.

## 4  FELDMAN & ZHANG (2020) LIMITATIONS

In the previous section we discuss how Feldman & Zhang (2020) define point memorization, approximate scores, and evaluate marginal utility. In this section, we explore critical limitations within each step and propose potential fixes.

### 4.1  IMPRECISE DEFINITION FOR MEMORIZATION:

The findings of Feldman & Zhang (2020), and their earlier theoretical work Feldman (2020), are based on the *premise* that Equation 1 provides a precise representation of memorized points (Section 3.1). We argue that the key limitation is not in their derived proofs, but in the use of an imprecise definition as the cornerstone of their work. Specifically, Equation 1 does not provide any threshold on how to differentiate memorized points from non-memorized ones. To further exacerbate the matter, points with high memorization scores (outliers) can behave in contradictory ways to ones with low memorization scores (large sub-populations). As a result, conclusions derived from one threshold can easily be disproved with points from a different threshold. The experiments based on this definition fail the falsifiability test; there simply is no way to prove your theory right or wrong.

Such contradictory results can be seen in the main result of the original work (and also corroborated by our own findings in the next section). The authors show that removing memorized points with high scores (i.e., using a high memorization threshold) does not impact test accuracy. However, removing memorized points with low scores does impact test accuracy. A natural question arises: which one of the two thresholds really represents the memorized data? Do we base our conclusion on the high thresholds and claim that memorization does not impact accuracy? Or based it on the low thresholds and argue the opposite?

### 4.2  HIGH APPROXIMATION ERROR:

To estimate the memorization scores, the authors provide an approximation algorithm (Section 3.2). In this subsection, we compare this approximation algorithm against the ideal LOO baseline. Specifically, we hypothesize *that the approximation algorithm over-estimates the memorization scores of small sub-populations*, making them indiscernible from singleton outliers. We find that this is caused by the approximation algorithm's sampling method producing biased partitions (i.e., ones that are dissimilar to the partitions created by LOO). As a result, the approximation algorithm has a high error for small populations, incorrectly marks them as memorized, resulting in many false positives. As a consequence, the original work reached an inaccurate conclusion about the role of memorization on model utility.

To illustrate this idea, consider our earlier hypothetical cat dataset containing 96 white, three black, and a single pink one. The approximation algorithm creates data shards by dropping a fraction of the points. This results in three types of dataset partitions. Partitions that contain 1) all three black cats, 2) some black cats, or 3) no black cats. On the other hand, LOO will only drop a single point at a time, and therefore, it's partitions will contain exactly two of the three cats *every* time. This means the approximation algorithm aggregates over partitions with fewer than two or no black cats, resulting in an overestimation of memorization scores. As a result, small sub-populations are most vulnerable to over-estimation.

In contrast, memorization scores of singleton outliers (pink cat) and large sub-populations (white cats) will not be impacted. In the case of outliers, this is because a biased partition is not likely because there is only one point. Similarly, because large sub-populations make such a significant fraction of the total data, there will always be a large number of them in any random partition.

**Setup and Methodology:** To evaluate our hypothesis, we compare the memorization scores of the approximation algorithm against LOO. Since executing LOO is computationally expensive, we perform our evaluation on CIFAR-10 using VGG-6 (Appendix A.1). We chose this setup because 1) the VGG-6 model trains fast (2 mins per model) allowing us to train the thousands of models required for the baseline LOO experiment and 2) the VGG-6 has a high test accuracy of 88%, indicating utility. We train VGG-6 using a batch size of 512, a momentum of 0.9, and a triangular learning rate scheduler, with a base rate of 0.4 and use the FFCV library (Leclerc et al., 2023) to speed up training. Having done so, we find the memorized points according to Equation 1.

Since running LOO on the entire dataset is not possible, we only execute it on a set of the 200 influential points, as they are used subsequently for evaluating marginal utility. We train a total of 20,000 models (100 models per point for a total of 200 points) for LOO assessment. And another 4,000 models for the approximation method.

**Results:** Figure 2 shows the error between LOO and the approximation method. We can see that the approximation method overestimates the scores for a large majority of the points. Specifically, only 8.5% of the points have a $< 5\%$ error in their memorization scores (from Equation 1). While the remaining points have errors as high as 57.5%. This means, if we set the memorization threshold of $\geq 25\%$, the approximation algorithm produced 16% false positives. The rate becomes even worrisome at higher memorization thresholds. For example, if we set the memorization threshold of $\geq 45\%$, the approximation algorithm produced 70% false positives. This means a significant fraction of points are incorrectly marked as memorized by the approximation algorithm.

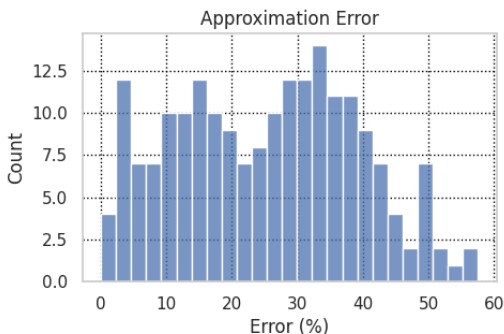

Figure 2: Error between the approximation method by Feldman & Zhang (2020) and the ideal leave-one-out.

This confirms the first part of our original hypothesis that the approximation algorithm overestimates the memorization scores.

For the second part of the hypothesis, our goal is to ascertain which subset of points are impacted the most by over-estimation (i.e., outliers or small sub-populations). To do so, we use the SPP outlier detection method Hendrycks & Gimpel (2016) which outputs a score between 1 and 0, with outlier points being scored closer to 0. We selected a 100 points that had the largest approximation error and a 100 points with the lowest. SPP score for the points with the lowest error was $0.32 \pm 0.26$ (indicating that these are outliers), while the points with the highest error have an average SPP score of $0.77 \pm 0.26$ (indicating that these are small sub-populations). This confirms the second part of our hypothesis: small sub-populations are most vulnerable to over-estimation.

### 4.3 FLAWED MARGINAL UTILITY:

After identifying the memorized points, the authors finally calculate their marginal utility (Section 3.3). This is where we observe a distribution shift error, a common oversight in machine learning research. Specifically, when calculating the utility of memorized points, the authors accidentally remove *all* the points above a specified threshold, thereby dropping *entire* sub-populations from the training data. This causes a complete or near complete sub-population purge from the training data, resulting in a distribution shift. This approach has shown to be sub-optimal Zhang et al. (2017b) because it results in a biased classifier. This is because the training data will no longer possess certain sub-populations that exist in the test data. This prevents the model from effectively learning the features of the removed sub-populations, and consequently, leads to poor model performance on the corresponding test points. Therefore, the drop in model accuracy observed by Feldman &

Zhang (2020) was *not* due to the marginal utility of the memorized points but due to the induced distribution shift (because entire sub-populations had been removed).

To make the contradiction apparent, if the authors had removed all the points below a specific threshold (i.e., remove all the points with low memorization scores belonging to large sub-population), we would also observe a drop in test accuracy. This would not mean that non-memorized points are necessary for generalization, but is a result of an induced distribution shift.

This idea can be understood using our earlier cat dataset example, presented in Section 2.1. If we remove all the black cats (three of them in total) from the dataset that contains another 94 white ones, the corresponding model will likely perform poorly on black cats in the test set. Similarly, if we remove all 94 white cats, the model will perform poorly on the white cats in the test set. Unfortunately, as was in the case of approximation error, small sub-populations are most vulnerable here as well.

## 5 RE-EVALUATING MARGINAL UTILITY

Having identified the issue of over-estimation, we now re-examine the relationship between marginal utility and memorization. Since the existing method of studying the relationship (Section 3.3) will exacerbate the above mentioned issues, we take a different approach. We study how memorization changes at different levels of generalization. We train 19 different combinations of models, datasets, and training parameters. We find that the memorization and generalization have a strong negative correlation (Pearson score 0.997) i.e., as one increases, the other decreases, thereby showing that memorization is not necessary for generalization.

### 5.1 SETUP

To expose the relationship between memorization and generalization, we train a series of models with an varying number of 1) trainable weights, 2) datasets, 3) architectures, and 4) training optimizations. By changing these variables, we are able to manipulate test accuracy and observe the change in memorization. Specifically, we designed four models using VGG blocks (VGG-0.5M[2], VGG-1M, VGG-8M, VGG-20M), two model Resnet models (Resnet-18 and Resnet-50), and two ViT models (Tiny and Small). We run our experiments over three datasets (CIFAR-10/100, Tiny ImageNet). Lastly, to isolate the impact of training procedure (i.e., weight decay and data augmentation), we repeat our CIFAR-10 experiments without these optimizations. Running such exhaustive experiments exposes the relationship between test accuracy and memorization. In all, this provides us with 19 data points of experiments to understand memorization and generalization.

We identify the memorized points using Equation 1. Similar to Feldman & Zhang (2020), we train the models for 100 epochs, using a batch size of 512, with a triangular learning rate of 0.4, weight decay of $10^{-5}$ for all models. We train 2,000 models for each of the models. In the case of ViT, we train 500 models using a learning rate of 0.1.

**Results** Figure 3 shows the results of our experiments. We can clearly observe the *strong* negative correlation (Pearson 0.997) between memorization and generalization (test accuracy): This means that as model utility increases, memorization decreases. We this trend hold across all models and datasets (i.e., both CIFAR-10/100 and Tiny ImageNet).

Similarly, we can observe that this observation holds even for models that trained with or without data-augmentation and weight decay. Even though these are common in modern training pipelines, we remove them to isolate their impact on memorization. In both cases (with and without these optimizations) we find that as model accuracy increases, memorization decreases.

The intuitive reason for this is that deeper models are better equipped to learn more meaningful features, which enables them to classify points correctly even if they are excluded from the training data. Test accuracy serves as an indicator of the quality of features learned by the model—higher accuracy implies better feature learning. Therefore, as deeper models learn higher-quality features, they tend to memorize fewer data points and generalize better to new data. The behavior of lower accuracy models further illustrates this: lacking the capacity to learn robust features, they are more

---

[2]Details of the architecture are provided in the Appendix

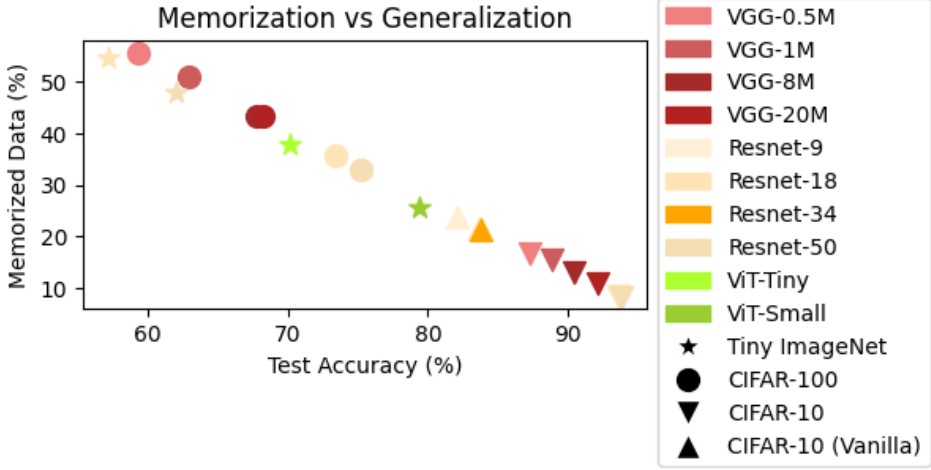

Figure 3: Memorization and generalization across different combinations of training procedures. We can observe a strong negative correlation between the two.

likely to misclassify points removed from the training data and thus rely more on memorization. Consequently, demonstrating that memorization is not necessary for generalization.

## 6 DISCUSSION

While Feldman & Zhang (2020) made valuable contributions, in the previous section, we showed that, after having accounted for the three limitations, memorization and generalization are strongly negatively correlated. The original incorrect results were a result of the high approximation errors (Section 4.2), distribution shifts (Section 4.3) and an imprecise definition (Section 4.1). As a result, the original authors incorrectly marked a significant number of points as memorized. However, by varying test set accuracy by iterating over different model architecutres, datasets, and training parameters, we find not only that memroization is not necessary for generalization, but infact is strongly negatively correlated.

Memorization has a direct implication for privacy research. This is because memorized points are vulnerable to membership-inference attacks (Carlini et al., 2022a). Feldman & Zhang (2020) created a tension between generalization and privacy. This is because they claimed that memorization was needed for generalization while other works demonstrated that memorization was harmful to privacy (Carlini et al., 2022a; Leino & Fredrikson, 2020; Carlini et al., 2019; Li et al., 2022). In other words, generalization and privacy can not be simultaneously achieved. While this might have dissuaded researchers in the community, our work shows that this tension does not exist. This is because memorization is not necessary for generalization. In light of these results, future researchers are encouraged to explore methods to build models that both generalize and are private.

## 7 RELATED WORK

One of the first papers to discover memorization deep learning models was Zhang et al. (2017a). They showed that models can fit completely unstructured images even if these consist of random Gaussian noise. Since then, there has been a tension between memorization and generalization and how they impact model performance (Chatterjee, 2018). Earlier works focused on limiting model memorization, thereby forcing the model to learn patterns instead. This was partly motivated by the fact that memorization exposed models to privacy risks (e.g., membership inference) (Carlini et al., 2019; 2022b). As a result, different methods were developed to counter memorization, which included using regularization (Arpit et al., 2017), filtering weak gradients (Zielinski et al., 2020; Chatterjee, 2020), adjusting model size (Arpit et al., 2017; Zhang et al., 2019). While these methods did reduce model memorization, they did so at the cost of model accuracy.

However, the true impact of memorization on model behavior was yet unknown. This first and foremost required methods to identify memorized points. A number of post-hoc methods were

developed to identify them. These included clustering (Stephenson et al., 2021), repurposed membership inference attacks (Carlini et al., 2022b), pseudo leave-one-out method (Feldman & Zhang, 2020). Having developed the ability to identify these points, the authors were now able to study their impact on model efficacy. As we describe in Section 3, Feldman & Zhang (2020) demonstrated that memorization was in fact necessary for model memorization. However, this conclusion was incorrect and was a by-product of a number of methodological errors. By accounting for these errors and rerunning their experiments, our results show that memorization has minimal impact on generalization.

## 8 CONCLUSION

Memorization is the ability of the model to fit labels to seemingly random samples. Recent work from Feldman & Zhang (2020) demonstrated that memorization is necessary for generalization. We show that the original work suffered from a number of crucial errors. These include the use of an imprecise definition, high approximation error, and distribution shift. In order to study the real impact of memorization, we train 19 different combinations of models, datasets, and training parameters. Having done so, we track memorization across generalization and find that memorization has a strong negative correlation with generalization. And therefore, memorization is not necessary for generalization.

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

# A  APPENDIX

Table 1: Symbols used and their meanings.

| Symbol | Meaning |
|--------|---------|
| $x_i$ | training data point |
| $y_i$ | training point label |
| $x_i'$ | test data point |
| $y_i'$ | test point label |
| $S$ | training set |
| $A$ | training algorithm |
| $n$ | size of the training set |
| $m$ | number of points removed from the training set |
| $h$ | trained model |
| $t$ | trial |

## A.1  MODEL ARCHITECTURES:

**VGG6 Architecture:** $64 \rightarrow MaxPool \rightarrow 64 \rightarrow MaxPool \rightarrow 64 \rightarrow MaxPool \rightarrow 64 \rightarrow MaxPool \rightarrow 512 \rightarrow MaxPool \rightarrow FC$.

