# OpenReview forum: "Is Memorization Actually Necessary for Generalization?"
_ICLR.cc/2025/Conference — Submitted to ICLR 2025_

### Official Review · Reviewer_LiN1 · 2024-10-28

**Soundness:** 1
**Presentation:** 2
**Contribution:** 2
**Rating:** 3
**Confidence:** 4

**Summary:**

The paper challenges Feldman & Zhang (2020)’s conclusion that memorization is necessary for generalization. The authors argue that there are three issues in Feldman & Zhang (2020)’s experiment design and propose a way to address these issues. They conducted experiments with their proposed changes on CIFAR-10/100 and present empirical results that contradict Feldman & Zhang (2020)’s claim.

**Strengths:**

The paper raises interesting points that challenge Feldman & Zhang (2020)’s conclusion. The explanations aid intuition, and the authors have also attempted to collect empirical evidence to support their claims. While I believe that rethinking and critically evaluating existing work is essential in the field and should be encouraged, I have a few concerns listed in the following section.

**Weaknesses:**

1. Lines 307-308 state, “It is clear that these small sub-populations are not memorized and therefore, should not be included in the set of memorized points.” This assertion seems overly confident. Is there rigorous evaluation on the exact number of small sub-population points that are or are not memorized? Are we entirely certain that none are memorized, and what direct evidence supports this? Could it be possible that some are memorized while others are not? Currently, this part lacks rigor.

2. The approach for identifying sub-populations also lacks rigor. The concept of a sub-population has no formal definition here, and the exact algorithm is unclear. For example, the phrase “check if multiple memorized points impact the same test set point; if so, they belong to the same sub-population” leaves questions—e.g., what quantitative measure of “impact the same test set point” is used? Additionally, in the second step (visual check), what criteria guide the decision that images belong to the same sub-population, and how do you justify the criteria you used? The granularity of the subgroups may also matter here. It is possible that, although A and B both impact the same test point based on the authors' criterion, their impacts are very different in magnitude—one is relatively large, while the other is small. In this case, removing A alone could significantly affect accuracy, whereas removing B might not. Grouping A and B together and removing both from the memorized set could therefore be problematic. Including a comprehensive list of identified groups in the appendix could be helpful, but more fundamentally, the definition of sub-populations here is highly subjective and does not seem scientific.
Using some synthetic data with clear definition of groups or existing datasets with sufficiently fine-grained group annotatations might strengthen the robustness of this analysis.

3. You mention that “small” sub-populations should not be identified as memorized. How do you quantify “small”? Specifically, after selecting the sub-populations in your experiments, how do you determine if they are small or large? And on what rationale is the threshold separating small and large based?

4. To support the claim that all small sub-populations are not memorized, it would be beneficial to provide the exact actual memorization scores for all points in the identified sub-populations. This would help confirm that none of them are indeed memorized. Additionally, you should also confirm that their memorization scores are overestimated by the approximation algorithm.

5. For a fair comparison, you could also apply the original method from Feldman & Zhang (2020) to your modified CIFAR-10/100 and demonstrate that their method produces a significant gap between the green and yellow curves. This would support your claim that their method falsely concludes that memorization is necessary.

6. How do you ensure that your modified CIFAR-10/100 datasets still follow a long-tailed distribution? Feldman & Zhang (2020) only makes their conclusion within the scope of long-tailed distribution datasets. I’ll expand on this in the next point.

7. Another important point concerns the fundamental conceptual understanding in this paper. Note that the conclusion from Feldman & Zhang (2020) is specifically in the context of long-tailed distributions, where there is a significant fraction of rare and atypical examples. They did not claim that memorization alone, isolated from other factors, is necessary for generalization; instead, they specifically associate their statement with long-tailed distributions. However, in this paper, the frequently used example, the hypothetical cat dataset, does not represent a long-tailed distribution, and using it may be misleading since it is not the scenario Feldman & Zhang (2020) address. A more appropriate example would be, for example, a setup with cats of 100 different colors, where three of the colors have 20 cats each, and the remaining 97 colors have only one or two cats each. In such a setting, it is clear that removing cats from those 97 colors would significantly impact generalization. Not memorizing those data points could be viewed as introducing an incurred or effective distribution shift, which does not contradict Feldman & Zhang (2020)’s conclusion. They did not suggest that the underlying distribution shifts do not contribute to accuracy drop; in fact, this is precisely the point in the context of long-tailed distributions.

**Questions:**

See the questions raised in the Weaknesses section.

---

> ### Author Response · Authors · 2024-11-27
>
> Dear reviewer,
>
> Thank you for your in depth feedback. Before we address your concerns, we would like to inform you that based on your concerns  (a lot of which were valid), we conducted additional experiments to further reinforce our findings. Our goal was to desgin an experiment that did not suffer from the weaknesses you pointed out.
>
> Specifically, instead of measuring change in accuracy after memorized points are removed, in Section 5 we track how memorization changes at different levels of generalization (test accuracy). We control model generalization by training 19 different combinations of models, datasets, and training optimizations. Specifically, we vary test accuracy by:
>
> - 1) changing model complexity (number of trainable weights)
> - 2) changing training optimizations (With and without weight decay and data augmentation)
> - 3) across different models (VGG, ResNet, and ViT) and
> - 4) across datasets (Cifar-10/100 and Tiny ImageNet).
>
> In each case, we train the model to the maximum test accuracy using the above listed parameters. Next, we record the generalization and memorization. Figure 3 shows the results of our experiments. We found that there is a **strong negative** correlation between generalization and memorization. In other words, as test accuracy increases, memorization decreases  (Perason score of -0.997). Our findings hold across different datasets, model architectures, training parameters, and model complexity, thereby strengthening our final conclusion.
>
> If Feldman et al’s claim was correct (i.e., memorization is necessary for generalization), we would have observed a positive correlation. However, we observe the opposite. This experiment provides one more piece of evidence to suggest that memorization is not necessary for generalization. We have updated the paper to reflect our findings. From the results of our experiments, we disprove the conclusion of the original work and show that memorization is not necessary for generalization

---

> ### Author Response · Authors · 2024-11-27
>
> > Lines 307-308 state, “It is clear that these small sub-populations are not memorized and therefore, should not be included in the set of memorized points.”
>
>
> We thank the reviewer for their response. Unfortunately, the only method to calculate the *exact( memorization scores, without creating false positives, is via the leave one out experiment. However, running it over 50,000 points by training 100 models for every removed point (i.e., training a total of 5x10^6 models) would be computationally intractable. However, we will be happy to soften the language so we are not making an overly confident assertion.
>
> > The approach for identifying sub-populations also lacks rigor…
>
>
> You are absolutely correct in your observations. In fact, motivated by your comments, we went back to our data and realized that sub-population identification method led to many false negatives. As a result, we opted for a different approach that we described at the beginning of the rebuttal.
>
> > You mention that “small” sub-populations should not be identified as memorized.
>
>
> Our goal in this part of the text was to explain which types of points are most vulnerable to approximation error by the original authors. In this case, “small” meant ones with higher than average SPP score. Though the reviewer is right to indicate that our language is imprecise. And we will be happy to clarify it in the text.
>
> > To support the claim that all small sub-populations are not memorized, it would be beneficial to provide the exact actual memorization scores for all points in the identified sub-populations.
>
> Calculating exact memorization scores of each point will require training 5x10^6 models, which is computationally intractable. However, our results indicate that images in Figure 1 suffer from the highest approximation error and as can be seen visually, demonstrate similar characteristics (demonstrating that they belong to the sub-populations).
>
> > Another important point concerns the fundamental conceptual understanding in this paper.
>
>
> The reviewer raises important points.  However, removing the singleton points (97 cats with different colors) from the tail will not impact generalization because **1)** The theoretical underpinnings of the Feldmen and Zhang’s experimental work, Feldman 2019 [1], argues that singletons are outliers and do impact accuracy. **2)** They also argue that memorized subpopulations impact accuracy (not outliers). So according to the original authors, the 97 singleton cats will be memorized, but will not impact accuracy. **3)** We can observe this in their original experiments: removing points from the tail (outliers/singletons using a high memorization threshold) did not impact accuracy. It was not until the authors started to reduce the memorization threshold to include more and more sub-populations, do we see a drop in accuracy.
>
>
> To address your concerns further, we ran the new experiments we described at the beginning of the rebuttal. Most of your criticism is valid. As a result, our goal was to design an experiment that does not suffer from any of the reviewer cited weaknesses.
> - 1. We do not make any over-confident assertions about which points are not memorized.
> - 2. We do not use an imprecise sub-population detection method.
> - 3. We do leverage “small” sub-populations, whose definition lacks rigor.
>
> Instead, we design an experiment where we merely vary the test accuracy and observe the corresponding change in model memorization. And results indicate that the two are negatively correlated. The original authors claim that “memorization is necessary for generalization”, which implies that there should be a positive correlation between the two. However, our results show that there is a **strong negative** correlation. As one increases, the other decreases.
>
> [1] https://dl.acm.org/doi/pdf/10.1145/3357713.3384290

---

> > ### Author Response · Authors · 2024-11-30
> >
> > Dear Reviewer,
> > As discussion deadline wraps up, please let us know if you have any more questions or suggestions on how we can improve our work.

---

### Official Review · Reviewer_tdvz · 2024-11-02

**Soundness:** 2
**Presentation:** 2
**Contribution:** 2
**Rating:** 3
**Confidence:** 3

**Summary:**

This paper revisits Feldman and Zhang's (2020) hypothesis that “memorization is necessary for generalization” by re-evaluating key elements in their approach. The authors point out several critical issues in the prior work's definitions of memorization score, the approximation algorithm, and the calculation of marginal utility. By addressing these limitations and introducing a modified experimental setup, this paper shows that memorization does not significantly contribute to generalization. Specifically, it highlights that Feldman and Zhang’s study overlooked the effects of sub-population shifts, which led to misleading conclusions.

**Strengths:**

- Perspective: The paper brings a valuable critical perspective to existing research, systematically reassessing established assumptions through a new experimental design. Its focus on sub-population shifts and errors in approximation algorithms is not only novel but also practically relevant.

- Empirical Contribution: The authors contribute by carefully analyzing the limitations in Feldman and Zhang’s original methods, particularly through data cleansing and detailed examination of approximation errors. The study effectively uses LOO (Leave-One-Out) experiments to reveal the conditions under which the original approximation algorithm yields incorrect results. While this approach is beneficial, more comprehensive experimental details would further strengthen this contribution.

**Weaknesses:**

- Concerns About Reproducibility: Although the paper provides valuable insights into the impact of sub-population shifts, it lacks detailed descriptions of the experimental setup, raising concerns about reproducibility. For instance, details on data set segmentation and specific threshold settings in Figure 2 and Table 1 are insufficiently clear, potentially making it difficult for other researchers to replicate the findings.

Presentation: Clearer explanations regarding the differences in the experimental setup in Section 4.2 and prior methods would improve comprehensibility. Demonstrating these differences through specific formulas would also clarify how definitions vary and the extent of the error introduced by each.

**Questions:**

- Regarding L278: "Since running LOO on the entire dataset is not possible, we only execute it on a set of the influential 500 points." Does this mean that the CIFAR-10 data was limited to 500 points?

- Settings in Figure 2 and Table 1: The description of how the training and test sets are divided in these experiments is unclear. To ensure reproducibility, could you clarify the specific methods of data division and the criteria for selecting data points?

- Accuracy Decline with Lower Memorization Thresholds: In Figure 3 and Table 1, accuracy declines as the memorization value threshold decreases. Could you provide further analysis on this point? Rather than dismissing prior findings outright, highlighting which conditions are significant and why this study clarifies them would enhance the reliability of the conclusions.

#### Additional Comments

- It would be helpful to provide an explanation of "SPP" in L300 for clarity.

- Finally, by sharing the data used for memorization and generalization analyses, this study could further support the research community by enhancing reproducibility and offering a reliable dataset for future work.

---

> ### Author Response · Authors · 2024-11-27
>
> Dear reviewer,
>
> Thank you for your in depth feedback. Before we address your concerns, we would like to inform you that based on the reviews, we conducted additional experiments to further reinforce our findings.
>
> Specifically, in Section 5, we track how memorization changes at different levels of generalization (test accuracy). We control model generalization by training 19 different combinations of models, datasets, and training optimizations. Specifically, we vary test accuracy by:
>
> - 1) changing model complexity (number of trainable weights)
> - 2) changing training optimizations (With and without weight decay and data augmentation)
> - 3) across different models (VGG, ResNet, and ViT) and
> - 4) across datasets (Cifar-10/100 and Tiny ImageNet).
>
> In each case, we train the model to the maximum test accuracy using the above listed parameters. Next, we record the rates of generalization and memorization for each model. Figure 3 shows the results of our experiments. We found that there is a **strong negative** correlation between generalization and memorization. In other words, as test accuracy increases, memorization decreases  (Perason score of -0.997). Our findings hold across different datasets, model architectures, training parameters, and model complexity, thereby strengthening our final conclusion.
>
> If Feldman et al’s claim was correct (i.e., memorization is necessary for generalization), we would have observed a positive correlation. However, we observe the opposite. This experiment provides one more piece of evidence to suggest that memorization is not necessary for generalization. We have updated the paper to reflect our findings. From the results of our experiments, we disprove the conclusion of the original work and show that memorization is not necessary for generalization.

---

> > ### Author Response · Authors · 2024-11-27
> >
> > Thank you for your feedback. We will be happy to go through our work to ensure every detail is present in order to replicate our findings. Specifically, our additional experiments (that we just described), should be fairly straightforward to replicate. We provide details on how to train the models in Section 5.
> >
> > It seems that your comments center around unclear text. We apologize for our oversight and will go back to add the necessary details. Unfortunately, we are unable to share the code and data due to legal restrictions. However, we will make the utmost effort to provide the necessary details on how to replicate our results.

---

> > > ### Author Response · Authors · 2024-11-30
> > >
> > > Dear Reviewer,
> > > As discussion deadline wraps up, please let us know if you have any more questions or suggestions on how we can improve our work.

---

> > > > ### Comment · Reviewer_tdvz · 2024-12-01
> > > > **Official Comment by Reviewer tdvz**
> > > >
> > > > Please read my initial question and I expect to your response to the questions I raised in my initial review.

---

> ### Author Response · Authors · 2024-12-01
>
> Apologies for our oversight. We address your concerns below:
>
> > Concerns About Reproducibility: Although the paper provides valuable insights into the impact of sub-population shifts, it lacks detailed descriptions of the experimental setup, raising concerns about reproducibility.
>
> Figure 2 shows the differences in *raw* memorization scores using Equation 1, without use of a specific thresholds. This was intentional as using an arbitrary thresholds might lead to inaccurate results (Section 4.1). We touch on this in Section 4.2 but will be happy to clarify this in the text.
>
> We used a threshold of 0.09 in table 2. However, due to the limitations of this experimental setup (as per the reviewers) we took a wholly different approach to calculate marginal utility (Section 5).
>
> > Presentation: Clearer explanations regarding the differences in the experimental setup in Section 4.2 and prior methods would improve comprehensibility.
>
> To enable a fair comparison, our experimental setup in Section 4.2 is nearly identical to the original work. We provide details in the text in paragraph 3. The only major difference is that we *also* run a Leave-one-out experiment. Here, we remove a single sample from the dataset, and retrain the model of the remaining points. Specifically, we are training on the CIFAR-10 dataset, which consists of 50,000 training points. We remove one point, and train a model on the remaining 49,999 points. We repeat this 100 times for each of the 200 points we evaluate (resulting in 20,000 models). And average the memorization score for each point using Equation 1. Both pLOO and LOO use the same equation to calculate the memorization score. The only difference is how the data is sampled (LOO: drop 1 sample at a time. pLOO: Drop random 30% of the data). We will make this clear in the camera ready version of the paper.
>
>
> > Regarding L278: "Since running LOO on the entire dataset is not possible, we only execute it on a set of the influential 500 points." Does this mean that the CIFAR-10 data was limited to 500 points?
>
>
> Thank you for this question. We meant that we used the full CIFAR-10 dataset, so the full 50,000 points. However the LOO procedure is computationally expensive to execute on the entire set of 50,000 points. Specifically, remove one point, retrain the model on the remaining 49,999 points, and repeat this 100 times. We can not run this on the entire dataset, as it will require training 100*50,000 models. Instead, we run the LOO procedure on the subset of points that have both high memorization scores AND have high influence. More details below in the response to the next question.

---

> > ### Author Response · Authors · 2024-12-01
> >
> > > Settings in Figure 2 and Table 1: The description of how the training and test sets are divided in these experiments is unclear. To ensure reproducibility, could you clarify the specific methods of data division and the criteria for selecting data points?
> >
> > We use the default training and test sets for Figure 2, which consists of 50,000 points. We select a subset of the most influential points for LOO i.e., points that have high memorization scores (>0.25) AND high influence scores (>0.15). These are the default values that were used in the original paper. We use the same one to ensure the most fair comparison. We selected these specific points as they are later used by the original authors for calculating marginal utility.  We will be happy to clarify this in the paper
> >
> >
> > Regarding Table 1, as mentioned earlier, due to concerns by the reviewers, we removed them from the paper. Instead, we opted for a more stable approach, described in Section 5.
> >
> > > Accuracy Decline with Lower Memorization Thresholds: In Figure 3 and Table 1, accuracy declines as the memorization value threshold decreases. Could you provide further analysis on this point? Rather than dismissing prior findings outright, highlighting which conditions are significant and why this study clarifies them would enhance the reliability of the conclusions.
> >
> >
> > Based on reviewer feedback, we replaced the original methodology (Figure 3 and Table 1) with a new method.
> >
> > Specifically, in Section 5, we track how memorization changes at different levels of generalization (test accuracy). We control model generalization by training 19 different combinations of models, datasets, and training optimizations. Specifically, we vary test accuracy by:
> >
> > - changing model complexity (number of trainable weights)
> > - changing training optimizations (With and without weight decay and data augmentation)
> > - across different models (VGG, ResNet, and ViT) and
> > - across datasets (Cifar-10/100 and Tiny ImageNet).
> >
> > In each case, we train the model to the maximum test accuracy using the above listed parameters. Next, we record the rates of generalization and memorization for each model. Figure 3 shows the results of our experiments. We found that there is a strong negative correlation between generalization and memorization. In other words, as test accuracy increases, memorization decreases (Perason score of -0.997). Our findings hold across different datasets, model architectures, training parameters, and model complexity, thereby strengthening our final conclusion.
> >
> > If Feldman et al’s claim was correct (i.e., memorization is necessary for generalization), we would have observed a positive correlation. However, we observe the opposite. This experiment provides one more piece of evidence to suggest that memorization is not necessary for generalization. We have updated the paper to reflect our findings. From the results of our experiments, we disprove the conclusion of the original work and show that memorization is not necessary for generalization.

---

> > > ### Author Response · Authors · 2024-12-02
> > >
> > > Dear Reviewer,
> > >  We hope we answered your questions and concerns. Please let us know if you have any other questions or suggestions.

---

### Official Review · Reviewer_AzVC · 2024-11-02

**Soundness:** 3
**Presentation:** 3
**Contribution:** 3
**Rating:** 6
**Confidence:** 3

**Summary:**

In this paper the authors examine and challenge the claim presented in Feldman & Zhang (2020) that memorization is necessary for generalization. They conclude that this claim is based on flaws w.r.t. the problem definition of memorization, the practice of estimating its intensity, and the analysis on its marginal utility. The authors further correct the correctable flaws and observe opposite results, concluding that memorization is not required for generalization.

**Strengths:**

This paper presents a well-supported opinion on the significant issue of memorization vs generalization. It's a solid rebuttal of a previous work. It is self-contained and very well-written.

**Weaknesses:**

As the authors point out in the paper, both their work and the previous one are basing their studies on the ambiguous and misquantified definition of memorization. Therefore both work suffer from the potential over-simplification and over-generalization of a complicated topic.

**Questions:**

1. Feldman & Zhang (2020) looks at memorization from the perspective of long tail study. One (my) way of interpreting their work is that memorizing the under-represented training points in a tail sub-population is necessary for the generalization within that sub-population at test time. The empirical study in this work does not disprove such interpretation. It would be interesting to know if the authors are opposing or confirming this particular point conceptually.

2. An alternative, and presumably more quantifiable definition of memorization could help further strengthen the claims in the paper.

---

> ### Author Response · Authors · 2024-11-25
>
> >didn't directly provide evidence if the "necessary condition" holds or not.
>
> We believe that these small sub-populations are not memorized, but are in fact learned by the model. Our evidence for this is two fold:
>
> **First**: In section 4, we show that the memorization approximation algorithms have high approximation error. Specifically, we find that small-subpopulations are marked as memorized by the approximation algorithm. However, running the expensive leave-one-out test (which is the gold-standard) shows that these points have much lower memorization scores (by as much as 57%)
>
> **Second**: Motivated by the reviews, we ran a additional experiments to compare model generalization and memorization (Section 5).
>
> Here, we track how memorization changes at different levels of generalization (test accuracy). We control final model generalization by training 19 different combinations of models, datasets, and training optimizations. Specifically, we vary test accuracy by:
>
> - 1) changing model complexity (number of trainable weights)
> - 2) changing training optimizations (With and without weight decay and data augmentation)
> - 3) across different models (VGG, ResNet, and ViT) and
> - 4) across datasets (Cifar-10/100 and Tiny ImageNet).
>
> From each experiment, we record the generalization and memorization. Figure 3 shows the results of our experiments. We found that there is a **strong negative** correlation between generalization and memorization. In other words, as test accuracy increases, memorization decreases  (Perason score of -0.997). If Feldman et al’s claim was correct (i.e., memorization is necessary for generalization), we would have observed a positive correlation. However, we observe the opposite. This experiment provides one more piece of evidence to suggest that memorization is not necessary for generalization. We have updated the paper to reflect our findings. From the results of our experiments, we disprove the conclusion of the original work and show that memorization is not necessary for generalization
>
>
> Both these experiments evidence that memorization is not necessary for generalization.
>
> > An alternative, and presumably more quantifiable definition of memorization could help further strengthen the claims in the paper.
>
> There are number of definitions of memorization available in current literature. Evaluating whether any of them actually fit the constraints is out of the scope of this work. However, we would be happy to point them out in the text.

---

> > ### Author Response · Authors · 2024-11-30
> >
> > Dear Reviewer,
> > As discussion deadline wraps up, please let us know if you have any more questions or suggestions on how we can improve our work.

---

### Official Review · Reviewer_38Zv · 2024-11-03

**Soundness:** 4
**Presentation:** 4
**Contribution:** 2
**Rating:** 5
**Confidence:** 3

**Summary:**

The paper reexamines the theoretical work of [Feldman, 2020] and [Feldman & Zhang, 2020]. They highlight three issues with the two papers: (i) arbitrary choice of memorization threshold makes certain claim from memorization not fallible at all; (ii) Approximation algorithm for memorization scores has high upward bias; (iii) unintended effect when removing data points.

Sec4, they provide concrete experimental results for the three results cited above.
Sec5, they redo the experiments based on the updated understanding and find the contribution of memorized points to test accuracy is exaggerated in the original paper.

**Strengths:**

* The experiments are technically solid, with setting and hyperparameters clearly stated.
* Lastly, the practice of revisiting an established paper and pointing technical inadequacy is good for the community.

**Weaknesses:**

* It's unclear what's the original contribution to the field the paper has. While pointing out the inadequacy in earlier work, the paper itself doesn't make the case for the role of memorization plays in generalization. It refutes the specific claims made by a particular set of papers, but didn't directly provide evidence if the "necessary condition" holds or not.
* The discussion of related work heavily focuses on the privacy-adjacent domain. There are other showing similar conclusion that does not have the problem mentioned by the author, e.g. [https://proceedings.mlr.press/v178/cheng22a/cheng22a.pdf] shows memorization is necessary for generalization without alluding to most of the concepts in Feldman et al. Similar, [https://proceedings.neurips.cc/paper_files/paper/2023/hash/bf0857cb9a41c73639f028a80301cdf0-Abstract-Conference.html] shows that explicitly enforcing memorization boosts the test time performance in actual experiments.

**Questions:**

The authors make a good lower-level technical discussion around Feldman et al., but how do those claims shed light into the memorization-generalization dilemma itself? This scientific question is independent of a particular set of papers that claim one way or another. Maybe the authors can discuss how their effort help clarify the underlying scientific question, not just pointing out lower-level technical flaws of earlier papers. I think this is potentially the true contribution of the experiments in the paper could shine.

---

> ### Author Response · Authors · 2024-11-25
>
> >didn't directly provide evidence if the "necessary condition" holds or not.
>
> We thank the reviewer for this comment. Motivated by one of their cited papers, we ran a new set of experiments to compare model generalization and memorization.
>
> Instead of measuring the drop in accuracy after removing memorized points, we track how memorization changes at different levels of generalization (test accuracy). We control final model generalization by training 19 different combinations of models, datasets, and training optimizations. Specifically, we vary test accuracy by:
>
> - 1) changing model complexity (number of trainable weights)
> - 2) changing training optimizations (With and without weight decay and data augmentation)
> - 3) across different models (VGG, ResNet, and ViT) and
> - 4) across datasets (Cifar-10/100 and Tiny ImageNet).
>
> From each experiment, we record the rates of generalization and memorization for each model.  Figure 3 shows the results of our experiments. We found that there is a **strong negative** correlation between generalization and memorization. In other words, as test accuracy increases, memorization decreases  (Perason score of -0.997). If Feldman et al’s claim was correct (i.e., memorization is necessary for generalization), we would have observed a positive correlation. However, we observe the opposite. This experiment provides one more piece of evidence to suggest that memorization is not necessary for generalization. We have updated the paper to reflect our findings. From the results of our experiments, we disprove the conclusion of the original work and show that memorization is not necessary for generalization
>
> > There are other showing similar conclusion that does not have the problem mentioned by the author
>
> The reviewer shared two important works. However, in both papers, we see the same issues in how memorization is defined.  For example, Yang et al do not check which of the points that they fit to K-NN meets the Feldmen et al’s definition.  Similarly, Chen et al use an arbitrary threshold in Equation 1 to bound train error (or degree of overfitting/memorization), which is similar to how Feldmen et al used random threshold to define memorization.
> In all three papers, the issue is not in how they run their experiments, but how they define memorization. A more meticulous approach like our 1) leave-one-out experiments (Section 4) and 2) our new memorization vs test accuracy experiments exposes the true underlying relationship between generalization and memorization.

---

> > ### Author Response · Authors · 2024-11-30
> >
> > Dear Reviewer,
> > As discussion deadline wraps up, please let us know if you have any more questions or suggestions on how we can improve our work.

---

### Official Review · Reviewer_fEpx · 2024-11-04

**Soundness:** 2
**Presentation:** 3
**Contribution:** 4
**Rating:** 5
**Confidence:** 4

**Summary:**

This manuscript reconsiders and critiques the key claim of Feldman and Zhang (2020), hereafter F&Z.  Three arguments are presented concerning methodological issues with F&Z:

(1) the definition of memorization includes an arbitrary threshold parameter, and it turns out that making changes to the parameter affect the conclusions.
(2) F&Z approximate the memorization score defined in (1) and this approximation has a bias that causes memorization to be overestimated.
(3) F&Z estimate the marginal utility of including an instance in the training set by a technique that overestimates the utility.

The authors conduct additional experiments to support their conclusions.

**Strengths:**

Due to the interest in F&Z---the article currently has 426 citations---undertaking this sort of investigation seems quite worthwhile, and such careful analyses of others' work is sadly uncommon in our field.

The paper is generally well written and clear.

**Weaknesses:**

Some claims and conclusions made by the authors either require additional support or are overly strong.

Some details of the work are unclear. Figure 2 took some work to decypher and unclear what threshold is being used.

My major concerns are as follows, with line numbers from the manuscript:

231: "The experiments based on this definition fail the falsifiability test".  This statement seems a bit strong. If one adopts their specific definition of memorization (i.e., a specific threshold), their theory is testable.  It just isn't terribly compelling.  As you argue, showing that the same result holds for a range of thresholds would strengthen the conclusion, even if it held only for that looser notion of threshold.

243: I don't follow your argument about removing memorized points with high vs. low scores.  The notion of changing the threshold (and removing points with scores above the threshold) does not correspond to these two cases, because the low-score case is removing ONLY points with a low score, not all points with a score above a low threshold. I'd suggest clarifying or removing this argument.

265: "subpopulations are most vulnerable to overestimation": This argument needs to be tightened up. If examples are dropped at random, the fraction of instances lying in a small population will be, statistically speaking, the same as the fraction of instances that lie in a large population. Maybe there's a higher variance argument, but I'm not clear. And maybe there's an argument that performance is a sublinear function of the number of members of a population:  the first one is essential, the second one is important, the third one less so, and so forth. If this argument were true, one would need to consider the absolute numbers of instances of a population that remain in the training set, not the fraction.

274: Since when is 88% a 'high test accuracy' for CIFAR10? What architecture were F&Z using?  At very least, shouldn't you acknowledge that they used a more powerful (and presumably closer to SOA) architecture, and the validity and significance of your results is conditioned on this difference.

282: If I understand, you train 20 models and look at the fraction of models that memorize according to some preset threshold. (I'm not clear on what that threshold is for Figure 2.) Doesn't this mean that your LOO estimate is only accurate to increments of 5% (1/20)? Again, I may be confused, but if you find 1 memorization in 20 and the approximation method produces .08, you'd call that an error of (.08-.05)/.05 = 60%. Even if your LOO method had produced 2 memorizations in 20, it would still have an error of 25%. The poor approximation method just can't win using this assessment. Might it not make more sense to look at the approximation error in absolute score, not relative score?

321: You claim that F&Z's decision to remove all points above a threshold (which must have been done not by accident but because it was too expensive to remove them one at a time, no?) causes a 'complete or near-complete subpopulation purge'. Is this claim based on your outlier scores (lines 300-306)? Whether your answer is yes or no, I would like to see more direct evidence (e.g., removed data points are classified correctly only if one of those removed data points is included in the training set).

399: "for most thresholds, the drop in accuracy is not statistically significant": What test have you done? Simply looking at overlap of +/- 1SD tells you little about significance. I would run an ANOVA to see if there was a main effect of item type removed (random vs. memorized) and an interaction between item type and memorization threshold. My bet is that you'll see an interaction for both CIFAR-10 and CIFAR-100, which indicates that whether or not  V&Z's claim holds true depends on the threshold. I believe that's a conclusion you wouldn't argue with, although it's not as strong as your "memorization is not necessary for generalization" claim (line 419).

Some minor issues:

68: "but due to common ML oversight". Don't you mean due to loss of the subpopulation?  The sentence refers to ML oversight twice.

72: "we disprove the conclusion of the original work..."  Seems fairer to say "we obtain an alternative or weaker conclusion of the original work..."

Figure 1: caption is ungrammatical

Jiang et al. (2021) emphasize the point the authors make about the role of subpopulations of various sizes and might well be cited.  See
https://proceedings.mlr.press/v139/jiang21k.html

244: "Fix" is a weird paragraph heading. Perhaps "Our solution" or "Our fix"?

251: should be "leave one out (LOO)" the first time you use acronym.

**Questions:**

Please address the comments I made in the 'weaknesses' section. I am open to raising my score, and I would like to see this work published in some form, but the arguments need to be tighter (or clearer, if it's my misunderstanding). In general, work of this sort requires converging evidence -- multiple experiments/analyses that drive home the same point. Any one bit of evidence may be weak (as I think some of your arguments are), but taken together they become compelling.

---

> ### Author Response · Authors · 2024-11-26
>
> Dear reviewer,
>
> Thank you for your in depth feedback. Before we address your concerns, we would like to inform you that based on the reviews, we conducted additional experiments to further reinforce our findings.
>
> Specifically, In Section 5, we track how memorization changes at different levels of generalization (test accuracy). We control model generalization by training 19 different combinations of models, datasets, and training optimizations. Specifically, we vary test accuracy by:
>
> - 1) changing model complexity (number of trainable weights)
> - 2) changing training optimizations (With and without weight decay and data augmentation)
> - 3) across different models (VGG, ResNet, and ViT) and
> - 4) across datasets (Cifar-10/100 and Tiny ImageNet).
>
> In each case, we train the model to the maximum test accuracy using the above listed parameters. Next, we record the generalization and memorization. Figure 3 shows the results of our experiments. We found that there is a **strong negative** correlation between generalization and memorization. In other words, as test accuracy increases, memorization decreases  (Perason score of -0.997). Our findings hold across different datasets, model architectures, training parameters, and model complexity, thereby strengthening our final conclusion.
>
> If Feldman et al’s claim was correct (i.e., memorization is necessary for generalization), we would have observed a positive correlation. However, we observe the opposite. This experiment provides one more piece of evidence to suggest that memorization is not necessary for generalization. We have updated the paper to reflect our findings. From the results of our experiments, we disprove the conclusion of the original work and show that memorization is not necessary for generalization

---

> > ### Author Response · Authors · 2024-11-26
> >
> > >231: "The experiments based on this definition fail the falsifiability test". This statement seems a bit strong.
> >
> > We shared this notion only so that the research community is 1) aware of the limitations of this specific definition 2) can propose a tighter definition as part of future work. We would be happy to soften the language.
> >
> > > 243: I don't follow your argument about removing memorized points with high vs. low scores.
> >
> > This is a good observation. We are trying convey to the reader that low score points behave differently than high score ones. The way the experiment was executed by the original authors, both low and high points were removed, entangling the behavior of the two types of points. However, this led to the conclusion that low and high points behave the same way. This is an incorrect conclusion because high score points consist of outliers (that do not have impact on accuracy) while low score points consist of small-subpopulations (and will reduce accuracy due to sub-population shift). We would be happy to clarify this in the text.
> >
> > > 265: "subpopulations are most vulnerable to overestimation":
> >
> > Thank you for helping us make this argument clearer. Yes, you are right, the argument based on absolute numbers is more compelling. We will clarify that in the text.
> >
> > > 274: Since when is 88% a 'high test accuracy' for CIFAR10?
> >
> > Thank you for this question. We meant that we wanted to train a model that had “high enough” accuracy for the model to be considered useful. While models can achieve significantly higher accuracy on CIFAR-10, they are much more expensive to train. As a result, it would not be possible to execute a LOO on them. We would be happy to acknowledge that we did not run the LOO experiment on large models like Resnet-50 (like done in the original paper) due to the computational burden (it would require training 20,000 models).
> >
> > > 282: If I understand, you train 20 models and look at the fraction of models that memorize according to some preset threshold.
> >
> > This is a fantastic point. To address it, we trained an additional set of models, increasing the pool from 20 to 100. Now, the LOO estimate is accurate within increments of 1% (1/100). We find that pLOO still underperforms LOO by almost the same margins (Figure 2).
> >
> > >  I would like to see more direct evidence (e.g., removed data points are classified correctly only if one of those removed data points is included in the training set).
> >
> > Thank you for this comment. Our argument is not “removed data points are classified correctly *only if one* of those removed data points is included in the training set”. Instead, it is “removed data points are not classified correctly if *none* of those removed data points (or sub-population) is missing in the training set.” This is widely accepted idea in ML and is also validated in practice (Santurkar et al 2020).
> >
> > > 399: "for most thresholds, the drop in accuracy is not statistically significant":
> >
> > This is a really good point. We ran this experiment, but found that p-values were not statistically significant for all thresholds. Upon closer inspection of the data, we realized our sub-population detection algorithm had too many false negatives. As a result of this finding, and reviewer feedback, we took the new approach (described in the beginning of the response). We track memorization vs generalization over 19 different combinations of models/datasets/parameters (Section 5). We find that memorization and generalization are negatively correlated (Pearson score 0.997, p-value 10^-21). The results in this case are statistically significant and provide more evidence to support our conclusion.
> >
> > > Minor
> >
> > We will be happy to fix all the minor corrections the reviewer pointed out.

---

> > > ### Comment · Reviewer_fEpx · 2024-11-26
> > >
> > > My thanks to the authors for their further efforts to clarify and improve their manuscript.
> > >
> > > I have to say, your new Figure 3 raises more questions to me than it answers. That spectacular correlation of .997 just seems unbelievable, especially considering it is based on multiple data sets with different difficulty and (implicit) noise levels. It would be a pretty spectacular (and unlikely) result to find the tight, linear trade off in Figure 3 to be a universal law of memorization/generalization across architectures/data sets/training procedures/etc.  I also don't know how to reconcile the result with the claim in the abstract: "We show that most memorization thresholds (the value that dictates whether a point is memorized or not) do not have a statistically significant impact on model accuracy."
> > >
> > > In any case, I resonate with a comment made by reviewer LiN1 that F&Z are focused on the setting of long-tailed distributions. Like reviewer LiN1, I'd like to understand how the setting studied by the authors relates.  I'm concerned that the claims made in the present work are too sweeping. See the cat example suggested by LiN1. A further, but silly example:  If I don't train a model, it won't generalize and it won't memorize. If I overtrain the model, it will generalize (more than an untrained model) but it may also memorize.  Might there be a sweet spot in the middle where a model will optimize generalization without showing any memorization? Sure, for some data sets, architectures, etc. But this fact does not invalidate the claims that F&Z make about long tail generalization.
> > >
> > > I do not follow one response to my review, specifically I'm having trouble parsing the following sentence; perhaps there's a typo:
> > >    _Instead, it is removed data points are not classified correctly if none of those removed data points (or sub-population) is missing in the training set._
> > >
> > > For now I am sticking with my review score, although I would still really like to see work such as this published, even if imperfect.  It has value for encouraging discussion and argumentation that advances the field.

---

> ### Author Response · Authors · 2024-11-27
>
> Apologies for the line the abstract. We will modify it to reflect the new experiments.
>
> Regarding the unlikely linear trade-off: We will be happy to concede in the text that this neat linear tradeoff might not exist for *all* models and *all* datasets. For that, we might need more data points and a wider study. However, we believe, this experiment represents 1) a strong indication that our understanding of memorization might be incomplete (if not flawed). 2) a first step towards helping the community realize that there are limitations in how we define memorization and calculate its value.
>
> Regarding the cat example: The reviewer described a long-tail consisting of 97 singleton points.  However, removing the singleton points (97 cats with different colors) from the tail will not impact generalization because **1)** The theoretical underpinnings of the Feldmen and Zhang’s experimental work, Feldman 2019 [1], argues that singletons are outliers and do not impact accuracy. **2)** They also argue that memorized subpopulations, not outliers, impact accuracy (Our work shows that these sub-populations are not memorized in the fist place). So according to the original authors, the 97 singleton cats will be memorized, but will not impact accuracy. **3)** We can observe this in their original experiments: removing points from the tail (outliers/singletons using a high memorization threshold) did not impact accuracy. It was not until the authors started to reduce the memorization threshold to include more and more sub-populations, do we see a drop in accuracy.
>
> Apologies. We meant: Points are misclassified by a model if they (or similar points) are not present in the dataset.
>
> We appreciate your feedback. If there is a way you feel we can tighten our arguments, please do not hesitate to let us know.
>
> [1] https://dl.acm.org/doi/pdf/10.1145/3357713.3384290

---

> > ### Author Response · Authors · 2024-11-30
> >
> > Dear Reviewer,
> > As discussion deadline wraps up, please let us know if you have any more questions or suggestions on how we can improve our work.

---

> > > ### Comment · Reviewer_fEpx · 2024-12-01
> > > **bottom line**
> > >
> > > I tried to step away from the details and distill the reason why I'm still not comfortable with the paper. It boils down to two things: (1) definitions:  In your last response to me, you distinguish between singletons/outliers and memorized subpopulations. I just don't see a qualitative difference between a single outlier instance (possibly repeated in the data set as F&Z identify) and a small population that lie far from the mean of a category. I encourage you to be more precise in your definitions and in your claims. (2) I appreciate the desire to answer your title question with a "no" to contrast with F&Z, but there's clearly different answers to different situations. The new experiment you report -- with a negative correlation between memorization and generalization -- is entirely consistent with the F&Z result that instances in the tail need to be memorized in order to obtain optimal generalization.  If a data set has mostly strong regularities, then indeed memorizing may reflect overfitting and generalization will be poor. If a data set has many small clusters of outliers, then generalization will be poor unless these clusters are memorized.
> > >
> > > I'm very sympathetic toward your effort at dissecting earlier work and understanding limitations on its claims. That's good scholarship and I want to see your paper published. But I don't think the current version of the paper offers sufficient clarity to improve our understanding.

---

> > > > ### Author Response · Authors · 2024-12-01
> > > >
> > > > We sincerely appreciate your help and support throughout the review process—thank you for your valuable contributions!
> > > >
> > > > > Critique 1:
> > > >
> > > > Our goal is to demonstrate that outliers are exclusively the points genuinely memorized, as we found using the baseline LOO procedure. However, errors in the approximation method and the imprecise definition result in sub-populations being incorrectly identified as memorized, entangling the conclusions for the two sets of points. Your concern is fair: our definitions do lack precision, which is part of our critique of F&Z. However, reasoning about their work using their own imprecise definitions, only to be criticized for imprecision, creates a classic catch-22. That transitions us to your next critique.
> > > >
> > > > > Critique 2:
> > > > To overcome this precise issue of imprecision, we employ the new experimental setup. Here, we make no assumptions about  the dataset, sub-population shifts, how the outliers and sub-populations are distributed, whether it has strong regularities, or clusters of outliers. What we find is that as memorization increases, accuracy decreases. *If* our experimental results were consistent with F&Z, then reducing memorization would *not* increase test accuracy. When clusters of points are memorized, we see a much lower test accuracy. It is not until these clusters are learned (that is low memorization score) that we see test accuracy improve. If the reviewer feels that the language of our conclusion is too strong (i.e., use of categorical negation of the original finding), we will be happy to soften it. However, the 1) negative trend between memorization and generalization, 2) imprecise definition for memorization, and 3) high approximation errors are hard to ignore.
> > > >
> > > > Once again, we appreciate the reviewer going through our responses and engaging in the discussion.

---

### Meta-Review · Area_Chair_TC1B · 2024-12-09

**Metareview:**

The paper revisits Feldman and Zhang (2020), where the author proposed a theory according to which a deep network that achieves high performance on natural images must memorise. This paper aims to confute the claims by Feldman and Zhang. Specifically, the authors identify three critical issues: 1. the definition of memorisation score; 2. the approximation algorithm used to evaluate the core; and 3. the calculation of marginal utility. By addressing these issues, the authors concluded that memorisation does not significantly contribute to generalisation.

**Additional Comments On Reviewer Discussion:**

This paper challenges an existing published result, which requires substantial evidence to avoid introducing further confusion into the literature. Unfortunately, the evidence provided was deemed insufficient by the reviewers.

The paper itself acknowledges ambiguities in certain definitions, which are central to critiquing the prior work. However, these ambiguities make it difficult to definitively refute the original thesis. Additionally, some results presented, particularly those added during the discussion phase, led to unclear conclusions and contributed to further confusion. Many of the claims were considered too strong, casting doubt on the overall thesis.
Reviewers also raised concerns about reproducibility, noting that the original submission under-reported key steps in the experiemtns. Furthermore, the authors revealed that the algorithm used in the initial submission for identifying sub-populations led to false positives, an issue only corrected during the rebuttal.

These points collectively weaken the paper's argument. The reviewers identified several flaws that make the paper unacceptable at this stage. I strongly encourage the authors to address these issues comprehensively and consider resubmitting once they are resolved.

---

### Decision · Program_Chairs · 2025-01-22

Reject